# Diurnal Oscillations of Fibrinolytic Parameters in Patients with Acute Myocardial Infarction and Their Relation to Platelet Reactivity: Preliminary Insights

**DOI:** 10.3390/jcm11237105

**Published:** 2022-11-30

**Authors:** Joanna Boinska, Marek Koziński, Michał Kasprzak, Michał Ziołkowski, Jacek Kubica, Danuta Rość

**Affiliations:** 1Department of Pathophysiology, Faculty of Pharmacy, Collegium Medicum in Bydgoszcz, Nicolaus Copernicus University in Toruń, 85-094 Bydgoszcz, Poland; 2Department of Cardiology and Internal Diseases, Institute of Maritime and Tropical Medicine, Faculty of Health Sciences, Medical University of Gdańsk, 81-519 Gdynia, Poland; 3Department of Cardiology and Internal Medicine, Faculty of Medicine, Collegium Medicum in Bydgoszcz, Nicolaus Copernicus University in Toruń, 85-094 Bydgoszcz, Poland; 4Department of Cardiology and Clinical Pharmacology, Faculty of Health Sciences, Collegium Medicum in Bydgoszcz, Nicolaus Copernicus University in Toruń, 85-168 Bydgoszcz, Poland

**Keywords:** acute myocardial infarction, fibrinolysis, tissue plasminogen activator, plasminogen activator inhibitor type-1, plasmin–antiplasmin complexes, α2-antiplasmin, dual antiplatelet therapy, clopidogrel, aspirin, circadian rhythms

## Abstract

There is limited information about diurnal changes in fibrinolysis parameters after acute myocardial infarction (AMI) and their relationship with on-treatment platelet reactivity. The aim of this study was to assess tissue plasminogen activator (t-PA), plasminogen activator inhibitor type-1 (PAI-1), α2-antiplasmin (α2-AP) activity, and plasmin–antiplasmin (PAP) complexes in 30 AMI patients taking dual antiplatelet therapy (DAPT), i.e., acetylsalicylic acid and clopidogrel. Fibrinolytic parameters were assessed at four time points (6 a.m., 10 a.m., 2 p.m., and 7 p.m.) on the third day after AMI using immunoenzymatic methods. Moreover, platelet reactivity was measured using multiple-electrode aggregometry, to assess potential differences in fibrinolytic parameters in low/high on-aspirin platelet reactivity and low/high on-clopidogrel platelet reactivity subgroups of patients. We detected significant diurnal oscillations in t-PA and PAI-1 levels in the whole study group. However, PAP complexes and α2-AP activity were similar at the analyzed time points. Our study reveals a potential impact of DAPT on the time course of fibrinolytic parameters, especially regarding clopidogrel. We suggest the presence of diurnal variations in t-PA and PAI-1 concentrations in AMI patients, with the highest levels midmorning, regardless of platelet reactivity. Significantly elevated levels of PAI-1 during the evening hours in clopidogrel-resistant patients may increase the risk of thrombosis.

## 1. Introduction

Many overlapping circadian rhythms, including heart rate, catecholamine levels, rest–activity exertion, posture, as well as emotional stress (i.e., cortisol levels), seem to be involved in the pathogenesis of acute myocardial infarction (AMI) in the morning hours [1,2]. Significant changes in fibrinolytic parameters have been proposed as other potential triggers of AMI [3,4,5]. Up-to-date studies indicate that any disturbances in the fibrinolytic system may contribute to the development of a serious hypercoagulable state. It has been proposed that modulators of fibrinolytic efficiency might be helpful in the secondary prevention and clinical outcomes of vascular diseases [6,7].

Fibrinolysis is a highly dynamic, enzymatic process that controls the conversion of plasminogen into the serine protease plasmin, which, in turn, dissolves fibrin [8]. The presence of fibrin initiates a conformational change in plasminogen, and its subsequent activation by specific plasminogen activators, such as tissue-type plasminogen activator (t-PA) and urokinase-type plasminogen activator (u-PA). t-PA, which is classified as a serine protease, is synthesized by endothelial cells and released into the circulation as a single-chain protein [9,10]. u-PA plays a significant role outside the vasculature via interaction with its cellular receptor. u-PA is produced by different cell types, i.e., epithelial cells, keratinocytes, and endothelial cells [10]. Plasminogen activator inhibitor-1 (PAI-1) is a serpin that regulates the fibrinolytic system through inhibition of t-PA and u-PA. Contrary to plasminogen activators, PAI-1 is produced by megakaryocytes and retained by platelets. However, smaller amounts of PAI-1 are also synthesized by other cells including endothelial cells, hepatocytes, and cardiomyocytes [11,12]. Alpha 2-antiplasmin (α2-AP) is a crucial inhibitor of plasmin. This glycoprotein is produced mainly by the liver and kidneys. However, smaller amounts of α2-AP have been detected in platelet alpha-granules. As a result of the α2-AP action, plasmin–antiplasmin (PAP) complexes form, which are accepted laboratory biomarkers of fibrinolysis [13,14].

Despite the growing attention paid to diurnal variations of fibrinolytic parameters, there is still limited information about changes in fibrinolysis parameters after myocardial infarction and factors that may affect their dynamics. The vast majority of studies that indicate the diurnal variability of fibrinolysis were published in the 1990s [15,16,17]. Moreover, many of those reports were based on studies of healthy volunteers, thus failing to reflect circadian variations of fibrinolytic parameters that occur in response to AMI [18,19]. Despite their exceptional value in contributing to our knowledge, these studies did not take current pharmacological strategies and non-pharmacological treatment options for AMI into account. Such strategies include dual antiplatelet therapy, which uses acetylsalicylic acid and a P2Y12 receptor inhibitor. The studies also neglected to mention the individual patient’s response to pharmacotherapy [20], which is valuable for us to study since high on-treatment platelet reactivity (HTPR) may affect the dynamics of changes in fibrinolytic factors, some of which are produced, stored, and released by platelets.

Against that background, the aim of this study was to assess various parameters of fibrinolysis, such as t-PA, PAI-1, α2-AP, and PAP complexes, at four time points (6 a.m., 10 a.m., 2 p.m., and 7 p.m.) on the third day after AMI. The paper also presents, in addition to the daily variability of these parameters, their relation to on-treatment platelet reactivity. It is still not known whether HTPR may affect the fibrinolytic parameters after AMI and thus possibly modify the clinical outcomes, meaning the presented research findings innovatively and significantly contribute to the research field.

## 2. Materials and Methods

### 2.1. Study Participants and Design

We designed and conducted a prospective cohort study of 30 consecutive patients treated with primary percutaneous coronary intervention (pPCI) for ST-segment elevation AMI at the Department of Cardiology and Internal Medicine of Dr. Antoni Jurasz University Hospital No. 1 in Bydgoszcz (Poland). The exclusion criteria were as previously described [5,21,22]. Importantly, they included any treatment with glycoprotein IIb/IIIa inhibitors, which might have skewed the results of this study.

On admission, all study participants received oral loading doses of clopidogrel and aspirin (600 and 300 mg, respectively), followed by maintenance doses administered at 8 a.m. (75 mg q.d. for both drugs). Immediately after hospital admission, all patients underwent pPCI. During pPCI, study participants received unfractionated heparin in a weight-adjusted manner (i.e., 100 U/kg) intra-arterially. No other anticoagulants were administered to the studied patients throughout the study period. Selected fibrinolytic parameters and platelet aggregation were assessed on the third day of hospitalization at four time points (6 a.m., 10 a.m., 2 p.m., and 7 p.m.).

We conducted this research following the previously described methodology [5]. We used widely accepted cutoffs for high on-treatment platelet reactivity (HTPR). In detail, high on-clopidogrel platelet reactivity was defined as platelet aggregation above 50 U in ADP tests. Similarly, high on-aspirin platelet reactivity was defined as platelet aggregation above 30 U in ASPI tests. Patients with low platelet aggregation in response to adenosine diphosphate (<50 U) and arachidonic acid (<30 U) were classified as clopidogrel- or aspirin-sensitive, respectively.

The study protocol and all procedures were approved by the Bioethical Committee of the Collegium Medicum in Bydgoszcz, Nicolaus Copernicus University in Toruń (KB599/2007).

### 2.2. Sample Preparation and Laboratory Tests

Blood was collected in two plastic tubes containing 3.2% sodium citrate (BD Vacutainer^®^, Becton Dickinson, Franklin Lakes, NJ, USA) and 25 µg/mL hirudin (Dynabyte, Munich, Germany) at selected time points on the third day of hospitalization. After centrifugation (at 2500× *g* at +4 °C for 20 min), plasma was stored at –80 °C until further use in assays.

t-PA concentrations in human plasma samples were measured using ASSERACHROM^®^ tPA (Diagnostica Stago, Asnières-sur-Seine, France). The detection limit of this assay was 1.5 ng/mL. The intra-assay coefficient of variation (CV) was between 5.1 and 7.3% and the inter-assay CV between 3.6 and 4.2%. PAI-1 concentrations were assessed using enzyme-linked immunosorbent assay kits (IMUBIND^®^ Plasma PAI-1 ELISA, American Diagnostica Inc., Stamford, CT, USA). The lower detection limit for PAI-1 was 2.2 ng/mL. According to the manufacturer’s instructions, intra- and inter-assay CVs for PAI-1 were 5.4–6.6% and 6.9–9.0%, respectively. PAP complexes in human plasma were measured using PAP micro-ELISAs (DRG Instruments GmbH, Marburg, Germany). The analytical sensitivity of the DRG ELISA was 10 µg/mL. The activity of α2-AP was determined using a coagulometer Coag-Chrom 3003 (BioKsel, Grudziadz, Poland) and Bio-Ksel plasma kits. Normal values for α2-AP activity were 80–120%.

### 2.3. Platelet Aggregation

Whole-blood multiple-electrode aggregometry (MEA) was performed within 2 h of blood sampling on a Multiplate Analyzer (Dynabyte Medical, Munich, Germany). ADP tests were used to detect a blockage of the platelet P2Y12 receptor (clopidogrel), and ASPI tests to detect an inhibition of the platelet cyclo-oxygenase (aspirin). For this reason, platelets were stimulated with two different agonists, ADP (6.5 µM) and arachidonic acid (15 mM), according to the manufacturer’s instructions. The mean values of the two independent determinations are expressed as the area under the curve of the aggregation tracing and displayed in arbitrary aggregation units (U).

### 2.4. Statistical Analysis

All statistical analyses were performed using Statistica version 13.3 for Windows (TIBCO Software Inc., Palo Alto, CA, USA). The Shapiro–Wilk test was used to test for normality of the continuous variables. Data are presented as the mean (±standard deviation) or median with interquartile range (IQR) as indicated, and as the count (percentage) for categorical variables. Diurnal variations of fibrinolytic parameters were analyzed with a non-parametric repeated measurements analysis of variance (Friedman’s test), followed by a post-hoc analysis with Dunn’s test. The chi-squared test was used to compare the distribution of frequencies. Due to lack of normally distributed data, the Mann–Whitney U test was used for comparing fibrinolytic parameters between divided subgroups at four time points. Correlations were tested with Spearman’s rank correlation test. The significance level was set at a *p*-value of less than 0.05.

## 3. Results

The study group (*n* = 30) reflected a standard population of patients with AMI referred for pPCI. The age of patients ranged from 43 to 82 years, with a median value of 57.5. Most of study participants were males and they frequently had comorbidities (e.g., hypertension and diabetes). The majority of patients suffered from overweight or obesity. Almost half of the studied patients were either current or past smokers. Detailed characteristics of the study group are presented in Table 1.

Among the thirty AMI patients, eight were resistant to aspirin and twelve to clopidogrel. Twenty-two patients were aspirin-sensitive and eighteen clopidogrel-sensitive. The analyzed subgroups did not differ in their clinical and laboratory parameters. However, we detected significantly high triglyceride concentrations in clopidogrel-resistant patients (*p* = 0.0292).

Statistical analysis conducted for the whole study group revealed a significant diurnal oscillation in t-PA and PAI-1 concentrations on the third day after AMI. As shown in Table 2, t-PA levels were significantly elevated at 6 a.m. (12.09 ng/mL) and 10 a.m. (12.05 ng/mL) as compared to 7 p.m. (10.26 ng/mL). Moreover, PAI-1 concentrations also decreased with time. The highest levels of this inhibitor were observed in the morning hours at 6 a.m. (8.95 ng/mL) and 10 a.m. (5.38 ng/mL) as compared to PAI-1 concentrations at other time points. PAP complexes were similar at the majority of time points. There were no differences in the activity of α2-AP between the analyzed time points.

Furthermore, in-depth analyses were conducted for aspirin- and clopidogrel-sensitive and -resistant subgroups to assess possible relationships between fibrinolytic parameters and the reactivity of platelets.

As shown in Figure 1, changes in the daily variations of t-PA and PAI-1 similar to those observed in the entire study group were detected.

Significant differences in t-PA were noted only in the subgroup of patients sensitive to aspirin (*p* < 0.001). PAI-1 diurnal variations were observed in both aspirin-sensitive and aspirin-resistant subgroups (*p* < 0.001; *p* < 0.001) (Figure 1). Surprisingly, significant time-course differences in PAP complexes were also observed within the aspirin-sensitive subgroup (*p* < 0.05). The lowest PAP concentrations were detected at 10 a.m. No significant daily changes were observed in α2-AP activity. Importantly, there were no differences in the concentration of the fibrinolytic parameters between the group of patients sensitive to aspirin and those resistant to aspirin at any of the four time points.

Moreover, significant diurnal variability of t-PA levels was detected in the subgroup of patients sensitive to clopidogrel (*p* < 0.05). In contrast, a similar relation was not observed for clopidogrel-resistant patients (*p* = 0.0751) (Figure 2). PAI-1 diurnal variations were observed in both clopidogrel-sensitive (*p* < 0.0001) and clopidogrel-resistant subgroups (*p* < 0.001). Furthermore, the difference in PAI-1 concentration between clopidogrel-sensitive (3.23 ng/mL) and clopidogrel-resistant patients (3.77 pg/mL) at 7 p.m. was statistically significant (*p* = 0.0210).

Contrary to t-PA and PAI-1 concentrations, the concentration of PAP complexes was significantly elevated at 7 p.m. (951.80 µg/L) in comparison to 6 a.m. (854.56 µg/L) and 10 a.m. (729.15 µg/L) only in the clopidogrel-sensitive subgroup of patients (*p* = 0.0331). There were no diurnal oscillations in α2-AP activity in the subgroups of different responses to clopidogrel.

No differences in t-PA, PAP complexes, and α2-AP activity parameters between clopidogrel-sensitive and clopidogrel-resistant patients were detected at any of the four time points.

In the entire study group, we observed strong correlations between t-PA and PAI-1 at 6 a.m. (R = 0.42; *p* = 0.0209), 10 a.m. (R = 0.45; *p* = 0.0121), and 2 p.m. (R = 0.61; *p* = 0.0003). There were also negative correlations between platelet aggregation in ASPI tests and t-PA concentrations at 10 a.m. (R = −0.45; *p* = 0.0108), as well as between platelet aggregation in ASPI tests and PAP at 2 p.m. (R = 0.36; *p* = 0.0466). Moreover, we detected a significant correlation between ADP-induced platelet aggregation and PAI-1 at 7 p.m. (R = 0.62; *p* = 0.002).

## 4. Discussion

The significant role of diurnal oscillations in affecting fibrinolysis parameters in the pathogenesis of AMI and during treatment is gaining importance. At the same time, previous studies have failed to take current pharmacological strategies into consideration as potential modifiers of fibrinolytic potential.

Against that background, we believe this is the first study to indicate diurnal oscillation of fibrinolytic parameters and the relationship with on-treatment platelet reactivity in AMI patients.

The present study demonstrates significant diurnal variations of t-PA concentration, with morning increases at 6 a.m. and 10 a.m. on the third day after AMI. Moreover, diurnal oscillations were also detected in its major inhibitor PAI-1. Significantly higher PAI-1 concentrations were observed at 6 a.m. and 10 a.m. compared to the afternoon hours. Moreover, we detected strong, direct correlations between t-PA and PAI-1 at three out of four analyzed time points.

As far as we know, there are limited data concerning t-PA concentrations post-AMI, originating only from single, observational studies. Similar findings were reported by Angelton et al., who also noted increased resting t-PA antigen and PAI-1 activity in the morning in patients with previous myocardial infarction (*n* = 15) [23]. Interestingly, in their study, t-PA activity was also increased in the evening. According to the authors, significant PAI-1 variations regulate the diurnal rhythm of fibrinolysis, and t-PA daily oscillations follow the PAI-1. The results of the present study partially support this hypothesis as we detected a relationship between t-PA and PAI-1 at three time points. However, noting that correlation does not imply causation, so we cannot fully support the idea of PAI-1 controlling the level of tPA in the regulation of fibrinolysis post-AMI. An observational study by Ganti et al. also revealed the highest t-PA and PAI-1 levels in the early morning (10 p.m.–4 a.m.), compared to 4 p.m.–10 p.m. [24]. Nevertheless, the small size of the sample (only three patients in the early-morning subgroup) prevents any generalizations to the wider population.

Furthermore, the vast majority of studies in healthy individuals revealed increased t-PA values in the evening. Bridges et al. observed elevated t-PA levels between 4 and 8 p.m. in young healthy volunteers [25]. The lowest concentration of t-PA was present in the morning. Budkowska et al. presented diurnal changes in the fibrinolytic system in 66 healthy volunteers. They observed the highest tPA levels in both sexes at 2 p.m. as compared to 8 a.m. [9]. Angelton et al. also reported a diurnal rhythm of tPA, with the highest tPA activity in the evening [23]. In contrast to the cited results, Rudnicka et al. detected increased t-PA levels in the morning, with the highest mean value at 10 a.m. (5.53 ng/mL) [26]. Research by Rudnicka et al. was performed on 9377 men and women aged 45 years on average, which enhances the credibility of their study.

Diurnal variations of PAI-1 levels after AMI were also studied by Angleton et al. and Ganti et al. [23,24]. The authors reported increased PAI activity in the morning compared to the evening. Unfortunately, we could not find any other studies in the available literature that assessed the diurnal fluctuations in PAI-1. Most of the research instead determined if and how treatment strategies may influence PAI-1 levels. Collet et al. measured PAI-1 levels in AMI patients at admission and 24 h later [27]. They reported a significant increase in this inhibitor, with the highest PAI-1 value a day after AMI. An interesting study by Rapold et al. reported that PAI-1, both the antigen and its activity, significantly increased in AMI patients treated with alteplase [28]. PAI-1 was not only high in the initial measurement (threefold higher than in healthy volunteers at 8 a.m.) but remained high and even increased, reaching the peak 12 h after thrombolytic treatment. Katsoros et al. determined PAI-1 before and 24 h after PCI with drug-eluting stent implantation [29]. They concluded that PAI-1 concentrations can help identify patients at increased risk of developing in-stent restenosis.

The results of our own research indicate increased activity of the fibrinolytic system after myocardial infarction, with the highest concentrations of t-PA and PAI-1 midmorning. Increases in both factors may reflect their mutual cooperation in the regulation of fibrinolysis. Diurnal oscillations in t-PA levels in AMI patients were the opposite of those observed in healthy volunteers. A potential mechanism behind this shift is a response to thrombotic risk. In the literature, there is a common agreement on circadian variation in the occurrence of AMI, with a peak incidence in the morning [30,31,32].

We did not find any diurnal oscillations of α2-AP and PAP complexes three days after AMI. To date, there are limited data on the diurnal rhythm of these fibrinolytic parameters. The results of this research are similar to those of Akiyama et al., who demonstrated a lack of any diurnal changes in α2-AP activity and PAP complexes in healthy adults [33].

In the next step, we focused on the mutual relations between fibrinolytic parameters and on-treatment platelet reactivity.

In the present study, significant diurnal changes of t-PA levels were only observed in the aspirin-sensitive subgroup of patients. However, we believe that the lack of significant differences in t-PA levels in aspirin-resistant patients is probably due to the low number of patients in this subgroup, since diurnal oscillations of PAI-1 levels were observed in both subgroups, regardless of on-aspirin platelet reactivity. Moreover, diurnal variations of the two parameters were analogous to those observed in the entire study group. To our surprise, we detected significant diurnal oscillations in PAP complexes in aspirin-sensitive patients, with the lowest PAP concentrations observed at 10 a.m. However, there were no differences in fibrinolytic parameters between aspirin-sensitive and aspirin-resistant patients at any of four time points

Aspirin impairs platelet aggregation via irreversible cyclo-oxygenase inhibition. However, aspirin’s properties go well beyond its antiplatelet effect. Experimental studies indicate that this drug may also impact fibrinolytic parameters, with the effect dose-dependent [34,35]. Levin et al. found that a high dose of aspirin (650 mg twice daily) did not change resting levels of t-PA, PAI-1, or PAP complexes, but decreased t-PA activity [34]. A multivariate analysis by Geppert et al. revealed that the chronic intake of acetylsalicylic acid (100 mg/d) was inversely related to tPA antigen levels [35]. A possible explanation for this relationship is that aspirin exerts anti-inflammatory actions and may modify t-PA release from endothelial cells. However, the results of our study do not present any differences in t-PA levels between aspirin-sensitive and aspirin-resistant patients. Accordingly, we propose that is still unclear whether aspirin may influence the synthesis or release of fibrinolytic factors.

To our knowledge, the present study is the first report demonstrating a potential modulation of fibrinolytic components in AMI patients by clopidogrel therapy. We found significant diurnal variations of t-PA, PAI-1, and PAP complexes in clopidogrel-sensitive patients. However, PAI-1 oscillations were also significant in patients with high on-clopidogrel platelet reactivity.

Notably, we also detected significantly higher PAI-1 levels at 7 p.m. in clopidogrel-resistant patients compared to clopidogrel-sensitive patients (*p* = 0.0210). Furthermore, we observed a significant correlation between PAI-1 levels and ADP-induced platelet aggregation at 7 p.m. These observations deserve special emphasis. Increased PAI-1 values may reduce the fibrinolytic potential in clopidogrel-resistant patients. PAI-1 regulates the initial phase of fibrinolysis by forming stable complexes with plasminogen activators (t-PA and u-PA) [11,12,36]. PAI-1 is produced by different cells, i.e., megakaryocytes, fibroblasts, monocytes, macrophages, and adipocytes. However, once synthesized, it is stored in platelet alpha-granules [37]. Clopidogrel selectively inhibits platelet aggregation induced by ADP. As a result, clopidogrel sustains platelets in a resting state in the circulation. Considering this, what mechanism occurs in patients with HTPR? We can only speculate that the lack of desired platelet inhibition may cause platelet activation and subsequent degranulation. We believe that this may be a mechanism behind the elevated PAI-1 levels in clopidogrel-resistant patients. Studies by Zhao et al. and Sakata et al. revealed that long-term antiplatelet therapy with clopidogrel reduces plasma PAI-1 levels in patients with ischemic stroke [38,39]. According to our study, this effect seems to apply to patients who are sensitive to treatment. Further studies are needed to investigate the possible link between PAI-1 and response to antiplatelet therapy.

The present study has some limitations that should be acknowledged. A small number of patients may not be sufficient to reveal all possible daily changes in key fibrinolytic parameters. For this reason, our findings should be considered hypothesis-generating, and require confirmation in a larger population. Furthermore, due to the observational nature of our study, the results may have been influenced by confounding factors. To date, the HTPR phenomenon has been studied intensively but using different methods [40,41]; for this reason, its importance and impact on the clinical course and outcome remain to be elucidated. This study was also underpowered concerning the impact of intra-coronary thrombus burden, left ventricular wall motion abnormalities, and the presence of post-infarct aneurysm on diurnal oscillations of fibrinolytic parameters. Finally, it would be interesting to conduct a similar study on patients treated with other antiplatelet drugs or anticoagulants (i.e., ticagrelor, prasugrel, or bivalirudin).

## 5. Conclusions

Our findings suggest the presence of diurnal variations in t-PA and PAI-1 concentrations in AMI patients, with the highest levels midmorning, regardless of the platelet reactivity. Significantly elevated levels of PAI-1 during the evening hours in clopidogrel-resistant patients may increase the risk of thrombosis. The influence of antiplatelet therapy on the parameters of fibrinolysis deserves appraisal and a greater research focus.

## Figures and Tables

**Figure 1 jcm-11-07105-f001:**
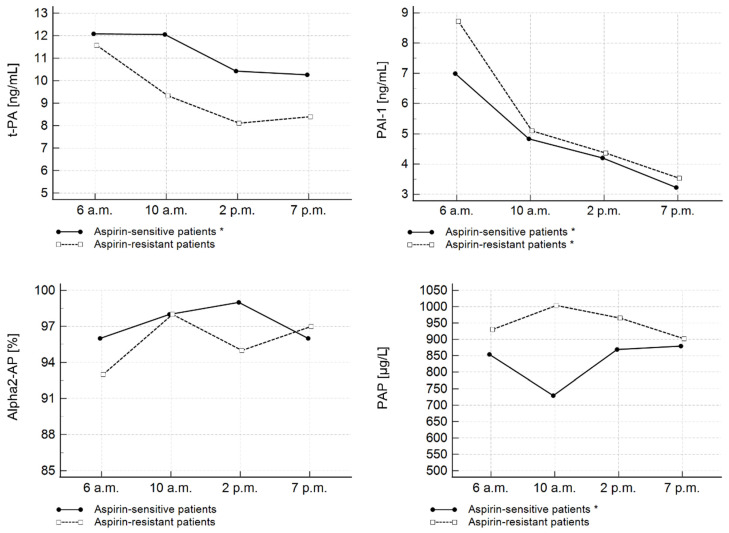
Diurnal changes of fibrinolytic parameters assessed on the 3rd day of hospitalization in AMI patients with high and low on-aspirin platelet reactivity. * Significant time-course differences within a group (*p* < 0.05). PAI-1—plasminogen activator inhibitor type 1; PAP—plasmin–antiplasmin complexes; t-PA—tissue plasminogen activator; α2-AP—alpha 2-antiplasmin.

**Figure 2 jcm-11-07105-f002:**
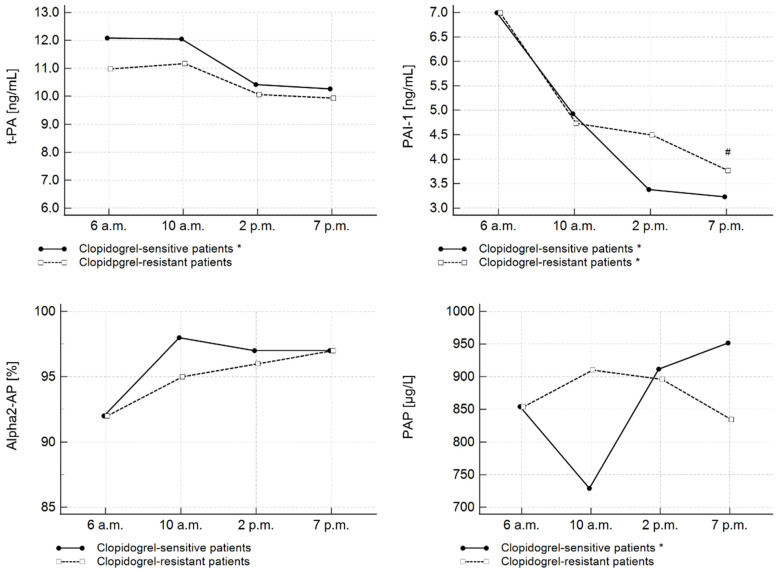
Diurnal changes of fibrinolytic parameters assessed on the 3rd day of hospitalization in AMI patients with high and low on-clopidogrel platelet reactivity. * Significant time-course differences within a group (*p* < 0.05); # significant difference in PAI-1 concentration between clopidogrel-resistant and clopidogrel-sensitive patients at 7 p.m. (*p* < 0.05). PAI-1—plasminogen activator inhibitor type 1; PAP—plasmin–antiplasmin complexes; t-PA—tissue plasminogen activator; α2-AP—alpha 2-antiplasmin.

**Table 1 jcm-11-07105-t001:** Patient characteristics.

		Entire Study Cohort		Aspirin-Sensitive Patients		Aspirin-Resistant Patients		*p*	Clopidogrel-Sensitive Patients		Clopidogrel-Resistant Patients		*p*
		*n* = 30		*n* = 22		*n* = 8			*n* = 18		*n* = 12		
		*n*	%	*n*	%	*n*	%		*n*	%	*n*	%	
Sex													
	female	11	37%	10	45%	1	12.5%	0.0976	7	39%	4	33.3%	0.7570
	male	19	63%	12	55%	7	87.5%		11	61%	8	66.7%	
History of smoking		14	47%	10	45%	5	62.5%	0.4089	9	50%	6	50%	1.000
Hypertension		22	73%	15	68%	6	75%	0.7185	11	61%	10	83%	0.1931
Diabetes mellitus		13	43%	9	41%	3	37.5%	0.8661	7	39%	5	42%	0.8790

		M	SD	M	SD	M	SD		M	SD	M	SD	

Age (years)		57.5	9.5	57.1	7.6	58.7	14	0.9257	55.2	9.4	61	8.9	0.0687
BMI (kg/m^2^)		28.1	4.2	27.7	3.73	29.1	5.4	0.6390	27.2	3.2	29.3	5.2	0.3302
Time from symptom onset (h)		5.1	4.4	4.9	2.6	5.6	7.6	0.2515	5.8	5.4	4.1	2.2	0.4924
CK-MB_max_ (U/L)		165	204	167	212	159	193	0.9625	184	219	136	184	0.8822
PLT (10^9^/L)		248	96	245	92	258	112	0.9252	225	84	283	106	0.0625
MPV (fL)		11.1	1.1	10.9	1	11.5	1.2	0.1971	11.1	1	11	1.2	0.7189
TCH (mg/dL)		210	46	214	47	201	4	0.4252	209	48	213	46	0.6566
HDL-CH (mg/dL)		39	12	40	13	35	9	0.3021	41	13	35	9	0.1275
LDL-CH (mg/dL)		149	40	150	40	145	4	0.7250	147	42	152	38	0.7508
TG (mg/dL)		108	56	110	56	104	59	0.7964	92	46	133	61	0.0292
HsCRP (mg/L)		12.9	10.9	12.0	10.6	16.8	14	0.6865	12.6	12.4	13.2	10	0.7928

BMI—body mass index; CK-MB—creatine kinase-myocardial band; HDL-CH—high-density lipoprotein cholesterol; HsCRP—high-sensitivity C-reactive protein; LDL-CH—low-density lipoprotein cholesterol; M—mean; MPV—mean platelet volume; PLT—platelets; SD—standard deviation; TCH—total cholesterol; TG—triglycerides.

**Table 2 jcm-11-07105-t002:** Diurnal changes of selected fibrinolytic parameters.

	3rd Day After AMI (*n* = 30)								Friedman’s ANOVA *p*-Value	Dunn’s Test *p*-Value
	6 a.m.		10 a.m.		2 p.m.		7 p.m.			
	Me	IQR	Me	IQR	Me	IQR	Me	IQR		
t-PA (ng/mL)	12.09	10.43–14.66	12.05	9.28–15.51	10.42	7.27–13.74	10.26	7.58–11.21	<0.001	1 vs. 4
										<0.05
										2 vs. 4
										<0.05
PAI-1 (ng/mL)	8.95	4.43–12.13	5.38	3.64–9.79	4.78	2.46–6.36	3.22	1.75–5.08	<0.001	1 vs. 3
										<0.05
										1 vs. 4
										<0.05
										2 vs. 4
										<0.05
α2-AP (%)	97	84–100	98	88–107	100	80–123	99	90–110	0.294	N/A
PAP (µg/L)	854.57	639.75–8938.63	817.47	639.75–1085.12	897.40	654.15–1055.82	855.06	693.04–1245.83	0.521	N/A

AMI—acute myocardial infarction; IQR—interquartile range; Me—median, N/A—not applicable; PAI-1—plasminogen activator inhibitor type 1; PAP—plasmin–antiplasmin complexes; t-PA—tissue plasminogen activator; α2-AP—alpha 2-antiplasmin.

## Data Availability

The data presented in this study are available in this article.

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
