# Peer review of "Diurnal Oscillations of Fibrinolytic Parameters in Patients with Acute Myocardial Infarction and Their Relation to Platelet Reactivity: Preliminary Insights"

_jcm, 2022, doi:10.3390/jcm11237105_

Round 1

Reviewer 1 Report

This article is very innovative in linking daily fluctuations in fibrinolytic parameters to platelet reactivity in patients with AMI. The results showed that the concentrations of t-PA and PAI-1 varied day and night in AMI patients, reaching the highest level at 10 am, which was not associated with platelet reactivity. Significantly elevated nighttime PAI-1 levels in clopidogrel resistant patients may increase the risk of thrombosis. I have the following suggestions:

1.      Is there a crossover problem when the patient is given two drugs? The fibrinolytic parameters were changed because of the combination of the two drugs?

2.      “We found strong correlations between t-PA and PAI-1 at 6 a.m. (R = 0.42, p = 0.0209), 10 a.m. (R = 0.45 p = 0.0121) and 2 p.m. (R = 0.61 p = 0.0003)”. The last paragraph in the results does not indicate which group?

3.      The last paragraph of the results does not show the correlation between platelet aggregation and fibrinolytic coefficient parameters.

Reviewer 2 Report

Dear authors; 

I have read the manuscript with great pleasure. It is reported that t-PA and PAI-I shows diurnal variation and their levels are higher in the morning contrary to the healthy people as reported in previous studies that their levels are usually higher afternoon. Moreover, some of the parameters had not diurnal variation in cases with aspirine and clopidogrel resistence that will guide future treatment targets for stent restenosis. 

I think this study is novel and educative however some revisions are required before further evaluation. 

- There are some misspellings (Line 70 -doeas, line 74- acetylosalycylic acid, line 328- resposne etc. ) that should be corrected. 

-During PCI and hospitalization patients should have been treated with antiocagulation. Did all of the cases undergo same antiocogulation protocol with the same agent? It should be reported if GpIIb/IIIa inhibitors were administered to any cases. Anticoagulation and use of other anti-thrombotic agents might affect the levels of fibrinolytic parameters. 

- Presence of wall motion abnormalities and aneurysms should be reported as it may trigger left ventricular blood stasis and thrombus formation that result in changes in thrombolytic parameters.  

-Classification of TIMI thrombus burden by angiographic images and evaluation of the relationship between coronary thrombus burden and diurnal variation in thrombolytic parameters would have added valuable information. 

Kind regards
